

# Dynamics of Majorana-based qubits operated with an array of tunable gates

Bela Bauer[1], Torsten Karzig[1*], Ryan V. Mishmash[2,3], Andrey E. Antipov[1]
and Jason Alicea[2,3]

**1** Station Q, Microsoft Research, Santa Barbara, California 93106, USA
**2** Department of Physics and Institute for Quantum Information and Matter, California Institute of Technology, Pasadena, CA 91125, USA
**3** Walter Burke Institute for Theoretical Physics, California Institute of Technology, Pasadena, CA 91125, USA

⋆ tokarzig@microsoft.com

## Abstract

We study the dynamics of Majorana zero modes that are shuttled via local tuning of the electrochemical potential in a superconducting wire. By performing time-dependent simulations of microscopic lattice models, we show that diabatic corrections associated with the moving Majorana modes are quantitatively captured by a simple Landau-Zener description. We further simulate a Rabi-oscillation protocol in a specific qubit design with four Majorana zero modes in a single wire and quantify constraints on the timescales for performing qubit operations in this setup. Our simulations utilize a Majorana representation of the system, which greatly simplifies simulations of superconductors at the mean-field level.

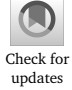

# 1 Introduction

Quantum computation holds great promise to solve some of the most challenging computational problems. Its progress, however, has been held back by the enormous difficulty of building reliable qubits, i.e., two-level quantum systems engineered to store and manipulate quantum information. While many qubit platforms have seen rapid progress in recent years and have enabled successful demonstrations of small but non-trivial quantum algorithms (see, for example, Refs. [1–3]), scaling these to the point where they allow robust error correction for a large number of qubits remains challenging.

Topological quantum computation [4] promises a giant leap forward by encoding quantum information into degrees of freedom that are inherently robust against external perturbations. This provides both a robust quantum memory as well as a discrete set of quantum gates that can be executed to high accuracy without further fine-tuning. Currently, the most promising hardware platforms to enable topological quantum computation are networks of one-dimensional topological superconductors that exhibit Majorana zero modes [5–12].

One of the earliest proposals for performing quantum computation with Majorana zero modes (MZMs) relies on physically moving these MZMs by tuning a series of electric gates under the superconductor in such a way as to drive different regions of the wire into the topological or non-topological regime [13]. MZMs will form at the boundaries between topological and non-topological regions, and as long as the separation between different MZMs is kept large enough and the changes in the electrochemical potential are performed sufficiently slowly [14–17], the manipulations of the state should be coherent in the low-energy subspace. We will refer to this as the 'piano key' approach to manipulating MZMs. Note that the requirement of sufficient separation and slow operations appear also in other methods of operating topological qubits [18–22]. At finite temperature, some of these restrictions may become more stringent [23, 24].

While theoretically appealing, this model of computation has been regarded as somewhat impractical due to the large voltages that might be required to tune the system in and out of the topological regime. This has led to a wide array of alternative means of manipulating MZMs [25–27]. However, the desire for a minimal Majorana-based qubit, together with potential improvements to gating techniques [28, 29], have led to renewed interest in qubit designs where MZMs are moved in this fashion. A possible design that is in principle able to demonstrate some of the advantageous properties of MZMs for quantum computation was proposed in Ref. [30]; if additionally operated at finite overall charging energy, it can also be seen as a minimal version of designs put forward in Ref. [27]. Similar ideas were pursued in Ref. [31].

In this paper, we first answer the question of how quickly MZMs can be moved using piano keys in the so far less-explored but more practical regime of sizable piano keys larger than the typical size of the MZMs. Tuning sizable regions through the topological phase transition introduces gap closing and reopening dynamics. We show that they are well-described by a Landau-Zener model and obtain a scaling relation for the diabatic errors, which we confirm in numerical simulations of the system. We then turn to simulations of simple protocols for Rabi oscillations in the qubit design of Ref. [30]. The simulations are performed both using Kitaev chains and more realistic models of spinful fermions on a lattice where the combined effects of induced superconductivity, Zeeman magnetic field, and spin-orbit coupling give rise to an effectively spinless $p$-wave superconductor. We confirm the scaling relation for diabatic errors in these more realistic settings, and furthermore discuss the constraints on the qubit operation timescales and the accuracy limitations due to the finite size of the qubit.

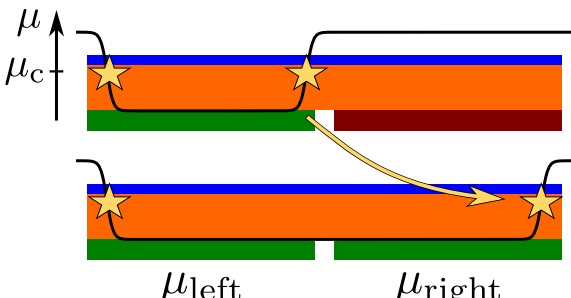

Figure 1: Piano-key move in a quantum wire (orange) proximitized by a superconducting shell (blue). The right MZM is moved by changing the electrochemical potential profile (indicated by solid black lines). The electrochemical potential on the left is held fixed in the topological regime; $\mu_{\text{left}} = \mu_{\text{top}}$, while on the right it is tuned from an initial value in the trivial regime to its final value in the topological regime, see Eq. (1). The tuning of $\mu_{\text{left}}$ and $\mu_{\text{right}}$ is performed by separate gates. The gate voltages are indicated by a green (dark red) color corresponding to the (non) topological regime.

## 2  Piano key dynamics

As a first step, we analyze the protocol illustrated in Fig. 1. We consider a one-dimensional superconductor whose electrochemical potential on the left and right halves can be independently tuned via electric gates. The gates are used to transport a MZM from the center of the system to its right end over a time scale $\tau$. More quantitatively, the electrochemical potential on the left half is fixed to a value $\mu_{\text{top}}$ corresponding to the topological regime, while the electrochemical potential on the right is tuned according to

$$\mu_{\text{right}}(t) = [1 - f(t/\tau)]\mu_{\text{triv}} + f(t/\tau)\mu_{\text{top}}. \tag{1}$$

Here $f(s)$ is a monotonically increasing function with $f(0) = 0$ and $f(1) = 1$; for example, one could choose $f(s) = \sin^2(s\pi/2)$, in which case $\partial_s f(s) = 0$ at $s = 0, 1$ as well. The right half thus begins in the trivial regime, but at time $\tau$ exhibits an electrochemical potential $\mu_{\text{top}}$ corresponding to the topological phase—thereby transporting the MZM as desired. In the strict adiabatic limit, the system is guaranteed to follow the instantaneous ground state throughout the protocol. In the remainder of this section we will explore both analytically and numerically diabatic corrections that arise when $\tau$ is finite. Our analytic approach closely follows earlier work by Damski and Zurek et al. [32, 33].

### 2.1  Analytical approach

For reasonably slow protocols, the diabatic corrections will be dominated by the point at which the gap of the instantaneous Hamiltonian is minimal. In our setup the gap is minimized when the piano key passes the critical point between the topological and trivial regimes, i.e., when $\mu_{\text{right}}(t) = \mu_{\text{c}}$. The finite size of the piano key will prevent a full closing of the gap at criticality and is therefore crucial for reaching the adiabatic regime. To obtain intuition, here we will estimate the probability for a diabatic transition out of the ground state by deriving a simplified two-level model for the system's low-energy spectrum near criticality and applying Landau-Zener theory.

The critical point corresponds to an Ising transition for which the low-energy degrees of freedom consist of right- and left-moving Majorana-fermion fields $\gamma_{R/L}$. For $\mu_{\text{right}}$ close to $\mu_{\text{c}}$

the right half of the superconductor is thus described by a low-energy Hamiltonian

$$\mathcal{H} = \int_0^{L_{\text{right}}} dx [-iv(\gamma_R \partial_x \gamma_R - \gamma_L \partial_x \gamma_L) + m(t) i \gamma_R \gamma_L]. \tag{2}$$

Here $L_{\text{right}}$ is the length of the piano key (right half of the system), $v$ is a velocity determined by microscopics, and $m(t)$ is a time-dependent 'mass' for the Majorana fermions tuned by the electrochemical potential. Near the transition we expect

$$m(t) = \lambda \, \delta\mu(t), \tag{3}$$

with $\delta\mu(t) = \mu_{\text{right}}(t) - \mu_c$ and $\lambda$ a model-dependent dimensionless coefficient. Additionally, the Majorana fields are subject to boundary conditions imposed by the left half of the system (which is topological) and the vacuum on the other end.

In an infinite system the instantaneous single-particle excitation energies are given by

$$\varepsilon(k) = \sqrt{(vk)^2 + (\lambda \, \delta\mu)^2}, \tag{4}$$

with $k$ the fermion momentum. Momentum is quantized to values $k_n$ in the finite-size piano key, however, yielding a discrete spectrum $\varepsilon_n \equiv \sqrt{(vk_n)^2 + (\lambda \, \delta\mu)^2}$ that we wish to now determine. The precise quantization condition follows from the boundary conditions noted above, and can be inferred using a variant of the arguments from Ref. [34]. Consider for the moment $\delta\mu = 0$. Two important properties arise here: First, in this limit we can equivalently repackage the right- and left-movers into a single *chiral* Majorana fermion defined on a system of length $2L_{\text{right}}$. (This chiral field simply corresponds to $\gamma_R$ on the interval 0 to $L_{\text{right}}$ and $\gamma_L$ on the interval $L_{\text{right}}$ to $2L_{\text{right}}$.) And second, the Majorana zero mode on the far left end of the system (see Fig. 1) must continue to have a partner on the right half, which at $\delta\mu = 0$ delocalizes across the critical region. The single chiral fermion must therefore exhibit *periodic* boundary conditions[1] with discrete momenta $k_n = \frac{2\pi n}{2L_{\text{right}}}$. Here the $n = 0$ level corresponds to the 'partner' Majorana zero mode while $n = 1, 2, \ldots$ correspond to finite-energy excitations.

We now focus on the ground state $|0\rangle$ and first excited state $|1\rangle$ within the same total-fermion-parity sector. The latter arises from the former by adding an excitation with wavevector $k_{n=1}$ *and* applying a Majorana-zero-mode operator to restore the original fermion parity; their instantaneous energy difference is then $\Delta E = \sqrt{\delta\varepsilon^2 + (\lambda \, \delta\mu)^2}$ with

$$\delta\varepsilon = \frac{\pi v}{L_{\text{right}}}. \tag{5}$$

This splitting is captured by an effective time-dependent Hamiltonian

$$H_{\text{eff}}(t) = \frac{1}{2}[\delta\varepsilon \, \sigma^x + \lambda \, \delta\mu(t)\sigma^z]. \tag{6}$$

In this form we can apply a standard Landau-Zener formula [35–38] to obtain the probability $p$ for exciting the system above the ground state:

$$p = \exp\left[-\frac{\pi}{2}\frac{\delta\varepsilon^2}{\lambda|\dot{\delta\mu}|}\right], \tag{7}$$

where $\dot{\delta\mu}$ denotes the time-derivative of $\delta\mu(t)$ evaluated at criticality. Next we specialize to

$$\delta\mu(t) = \left[1 - 2\sin^2\left(\frac{\pi t}{2\tau}\right)\right](\mu_c - \mu_{\text{top}}). \tag{8}$$

---

[1]By contrast, anti-periodic boundary conditions arose in Ref. [34]. In the setup addressed there the critical region was bordered on both sides by trivial phases, thus precluding the appearance of a Majorana zero mode at criticality.

This choice corresponds to Eq. (1) with $\mu_{\text{triv}} = 2\mu_{\text{c}} - \mu_{\text{top}}$, where the critical point arises at $t = \tau/2$, midway through the protocol. Hence $|\dot{\delta\mu}| = \pi|\mu_{\text{c}} - \mu_{\text{top}}|/\tau$, yielding a probability

$$p = e^{-\tau/\tau_0}, \quad \tau_0 = 2\lambda |\mu_{\text{c}} - \mu_{\text{top}}| \left(\frac{L_{\text{right}}}{\pi v}\right)^2. \tag{9}$$

Protocol times exceeding $\tau_0$ approximate the adiabatic limit.

It remains to determine the velocity $v$ and coefficient $\lambda$, which one can readily extract from a given microscopic model by examining the energy spectrum near criticality and fitting to Eq. (4). We will consider two specific microscopic realizations. The first is the Kitaev chain [5] with Hamiltonian

$$H_{\text{K}} = \sum_i \left[ -\mu c_i^{\dagger} c_i - \frac{w}{2} \left( c_i^{\dagger} c_{i+1} + H.c. \right) \right] + \frac{\Delta}{2} \sum_i (c_i c_{i+1} + H.c.). \tag{10}$$

At this point $H$ above is not intended to describe the entire superconductor from Fig. 1. Instead we consider a simple uniform chain, which suffices for extracting $v, \lambda$. The single-particle excitation spectrum is given by

$$\varepsilon(k) = \sqrt{[\Delta \sin(ka)]^2 + [w \cos(ka) + \mu]^2}, \tag{11}$$

where $a$ is the lattice spacing. Focusing on the critical point at electrochemical potential $\mu_{\text{c}}$ and expanding for small $k$ yields $v = a\Delta$ and $\lambda = 1$. In the Kitaev-chain realization, the characteristic time scale in Eq. (9) thus reduces to

$$\tau_0^{\text{K}} = 2|\mu_{\text{c}} - \mu_{\text{top}}| \left(\frac{L_{\text{right}}}{\pi a \Delta}\right)^2. \tag{12}$$

For a more realistic setting, we consider the canonical model of a quantum wire that exhibits topological superconductivity [7,8] due to a combination of spin-orbit coupling $\alpha$, Zeeman field $V_z$, and proximity-induced pairing $\Delta$:

$$H_{\text{QW}} = \int dx \, \psi^{\dagger} \left( -\frac{\partial_x^2}{2m} - \mu - i\alpha\sigma^y \partial_x + V_z \sigma^z \right) \psi + \int dx \, \Delta(\psi_{\uparrow}\psi_{\downarrow} + H.c.). \tag{13}$$

We give the lattice version of the above Hamiltonian, which we use in the numerical simulations, in Appendix A. The topological phase forms in the parameter regime $V_z^2 > \Delta^2 + \mu^2$, so that critical points occur at both $\mu_{\text{c}} = +\sqrt{V_z^2 - \Delta^2}$ and $\mu_{\text{c}} = -\sqrt{V_z^2 - \Delta^2}$. Two critical points arise because one can destroy the topological phase either by raising the density to enter a conventional superconducting state with two partially occupied bands, or by depleting the bands altogether, yielding a trivial strong-pairing phase. Our calculation applies to both cases, and we will include examples of both in our numerical simulations. Expanding the energy spectrum near criticality as above now gives $v = \alpha\Delta/V_z$ and $\lambda = \sqrt{1 - \Delta^2/V_z^2}$. The characteristic time scale in Eq. (9) is then

$$\tau_0^{\text{QW}} = 2\sqrt{1 - \frac{\Delta^2}{V_z^2}} |\mu_{\text{c}} - \mu_{\text{top}}| \left(\frac{L_{\text{right}}V_z}{\pi\alpha\Delta}\right)^2. \tag{14}$$

for the quantum-wire realization. Notice that as $V_z$ approaches $\Delta$ from above, $\lambda$ vanishes and thus so does $\tau_0^{\text{QW}}$, signaling a breakdown of our formalism. In this singular limit, the two phase transitions mentioned above coincide, i.e., the topological phase shrinks to a point. Our protocol thus requires $V_z > \Delta$.



Assuming $V_z = 0.3$ meV, $\Delta = 0.1$ meV, $L_{\text{right}} = 0.5\,\mu$m, $\alpha = 0.01$ eV·nm, and $|\mu_c - \mu_{\text{top}}| = 0.5$ meV, we find $\tau_0^{\text{QW}} \sim 10$ ns. Comparing this estimate to other schemes [18–22] where the relevant adiabatic timescale is typically the inverse topological gap $\sim 0.1$ns, we observe that large piano keys, corresponding to a small level spacing at the critical point, indeed lead to more stringent requirements for adiabaticity. We note, however, that some caution is warranted with the specific value of the above estimate for $\tau_0^{\text{QW}}$. Physical parameters such as the $g$-factor and the strength of spin-orbit coupling are generally renormalized from the values for an isolated semiconducting wire [39], which in turn can influence the level spacing [40]. Such effects can be particularly dramatic when the wire strongly couples to the superconductor since the relevant low-energy wavefunctions have significant weight also in the superconductor [41, 42]. Furthermore, the spin-orbit-coupling strength may be strongly affected by geometric effects due to confinement of the wavefunctions.

## 2.2 Numerical confirmation

To confirm the analytic estimates, we perform numerical simulations of the dynamical protocol using a Majorana representation of the Hamiltonian. While such a representation is in principle possible for any quadratic fermionic system, it is particularly suited to the simulation of the dynamics of superconducting systems at the mean-field level, where it greatly simplifies the computation of physical observables such as the joint parity of a collection of Majorana modes. For details of the numerical approach, we refer to Appendix B. We use the inter-site hopping as the energy unit, and the distance between the neighboring sites as the unit of length.

We simulate the protocol in Kitaev chains of lengths ranging from $L = 60$ to $L = 100$ (here $L_{\text{right}} = L/2$), and for various values of $\Delta$ and the initial and final electrochemical potential. We estimate the diabatic errors in two ways: First, by computing the overlap of the final wavefunction $|\psi_f\rangle$ with the instantaneous ground state of the final Hamiltonian $|\psi_0\rangle$ in the same parity sector as the initial state; the error is then $1 - |\langle\psi_f|\psi_0\rangle|^2$. Note that since there are at all times only two MZMs, there is no topological degeneracy in a fixed parity sector in this system. In a second, complementary approach, we compute the occupation of the low-energy subspace spanned by the two Majorana modes $\gamma_{1,2}$ which are defined through instantaneous single-particle eigenstates of the final Hamiltonian. Here the error is defined

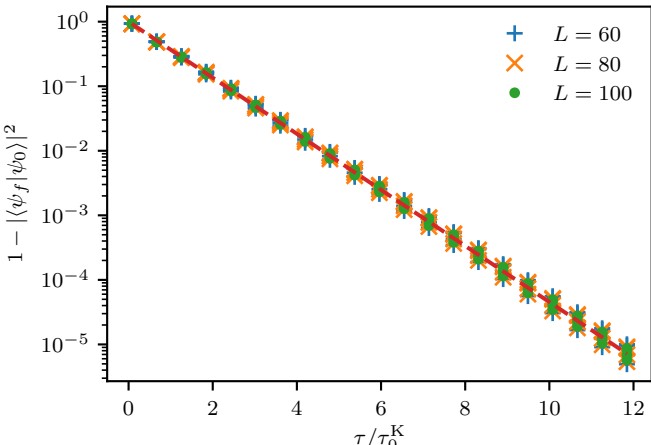

Figure 2: Numerical results for diabatic errors in a piano-key move simulated in a Kitaev-chain model. The data represents an array of parameters for $L = 60, 80, 100$, $\Delta = 0.3, 0.5$, and $\mu_{\text{triv}} - \mu_{\text{top}} = 0.5, 1.0$. The ramp shape is $f(s) = \sin^2(s\pi/2)$. The dashed line indicates a fit to $p = e^{-\tau/\tau_0^{\text{K}}}$; see Eq. (12).

as $1-(\langle i\gamma_1\gamma_2\rangle+1)/2$, where we use a convention such that $\langle i\gamma_1\gamma_2\rangle=+1$ in the ideal limit. This error measure is more relevant for applications to topological quantum computation as it specifically quantifies the weight of excitations from the Majorana wavefunctions, i.e., the low-energy single-particle operators that generate the ground-state manifold.

Figure 2 shows our numerical results for the overlap; with the exception of all but the fastest of protocols ($\tau \lesssim \tau_0$), the two error measures behave essentially identically. We find excellent agreement with the theoretical prediction over a wide range of protocols speeds and parameter choices. This collapse is good evidence that the simple Landau-Zener form considered above holds also in the more realistic setting where a continuum of states collapses at the critical point, as first observed in Refs. [32,33] in the context of the Ising model. For smaller systems, subleading finite-size corrections are more prevalent, and the results deviate more from the theoretical prediction. We note that for non-analytic choices of $f(s)$ the exponential behavior eventually crosses over to a power law (see, e.g., [19,20]). Due to the choice of a smooth first derivative of $f(s)$ and the large ratio between the gap at the point of non-analyticity $\sim \Delta\mu$ and the gap $\delta\varepsilon$ that is controlling the Landau-Zener dynamics, the prefactor of the power law is too small to be observed in Fig. 2. We checked that for a discontinuous first derivative $f(s)=s$ (for $0<s<1$) and small system sizes, the power law becomes observable at large times.

We perform similar simulations of the spinful model, Eq. (13). Figure 3, which is analogous to Fig. 2, shows our results for this case. Here we consider total system sizes between $L=100$ and $L=220$, and an array of values for the parameters $\Delta$, $V_z$ and $\alpha$ as listed in the caption of Fig. 3. (Recall that phase transitions occur at $\mu_c=\pm\sqrt{V_z^2-\Delta^2}$; hence we only examine combinations of the parameters with $V_z>\Delta$, where the topological phase has a finite extent.) The values for the electrochemical potential in the trivial and topological phases are chosen symmetrically around the critical value such that the phase transition occurs halfway through the evolution. We again consider both error quantities and find excellent agreement between them for all but very fast protocols.

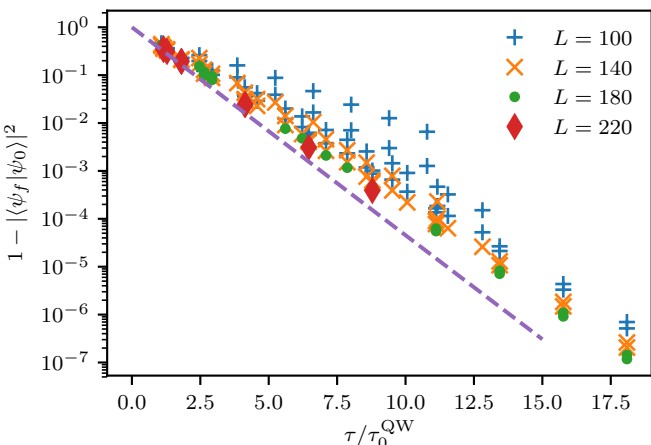

Figure 3: Diabatic errors in the piano-key protocol simulated using a spinful model for various combinations of $\Delta = 0.3, 0.5$, $V_z = 0.4, 0.6, 0.8$, $\alpha = 0.3, 0.6$, with the initial and final electrochemical potential appropriately chosen deep in the topological and trivial regimes, respectively. Total system sizes are $L = 100, 140, 180, 220$. Finite-size effects are more pronounced compared to Fig. 2 since the coherence lengths are much larger in the cases shown here. The dashed line indicates a fit to $p = e^{-\tau/\tau_0^{\mathrm{QW}}}$; see Eq. (14).

Compared to results for the Kitaev model, the data shown in Fig. 3 scatters much more

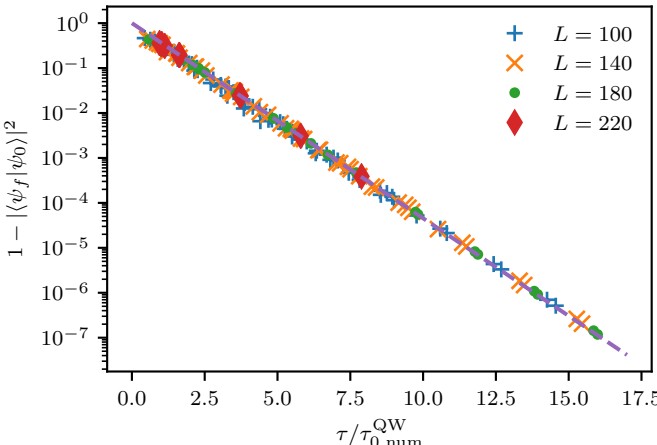

Figure 4: Diabatic errors in the same setup as Fig. 3, but now plotted as a function of $\tau/\tau_{0,\mathrm{num}}^{\mathrm{QW}}$. Equation (15) defines the characteristic time scale $\tau_{0,\mathrm{num}}^{\mathrm{QW}}$, which accounts for finite-size corrections to the size of the minimal excitation gap and the electrochemical potential at which it arises. The dashed line indicates a fit to $p = e^{-\tau/\tau_{0,\mathrm{num}}^{\mathrm{QW}}}$. The excellent data collapse evident here indicates that the scatter in Fig. 3 originates from finite-size effects, and that a simple Landau-Zener description continues to adequately capture diabatic errors in the spinful model.

around the predicted value. Close examination, however, shows that the deviation decreases for larger system sizes, and thus manifests finite-size effects. These corrections are more pronounced here compared to the case of a Kitaev chain since the coherence length is much larger for the parameters chosen in our simulations of the spinful quantum wire. Specifically, the minimal spectral gap of the finite-size lattice model along the Hamiltonian path, $\delta\varepsilon_{\mathrm{num}}$, generally differs from the value $\delta\varepsilon$ estimated in Eq. (5), and moreover can occur at an electrochemical potential that is shifted slightly away from $\mu_c$. We can find the value of $\delta\varepsilon_{\mathrm{num}}$ and the time at which it occurs for the family of instantaneous Hamiltonians using standard numerical techniques. Straightforwardly modifying Eq. (14) to incorporate these effects produces a more reliable estimate for the characteristic time scale in a finite-size system,

$$\tau_{0,\mathrm{num}}^{\mathrm{QW}} = \frac{4}{\pi}\delta\varepsilon_{\mathrm{num}}^{-2}\sqrt{1 - \frac{\Delta^2}{V_z^2}}|\mu_{\mathrm{c}} - \mu_{\mathrm{top}}|\frac{\partial f}{\partial s}(s_c), \tag{15}$$

where $s_c$ corresponds to the rescaled time $t/\tau$ where the minimal gap occurs, and $\frac{\partial f}{\partial s}(s) = (\pi/2)\sin(\pi s)$. Plotting the errors against $\tau/\tau_{0,\mathrm{num}}^{\mathrm{QW}}$ indeed generates excellent data collapse as seen in Fig. 4.

## 3 Qubit operations

We now turn our attention to qubit operations. We will examine a simple topological-qubit design [30], sketched in Fig. 5, consisting of a single quantum wire that can be partitioned into topological and non-topological regions by locally tuning the electrochemical potential using five gates. As shown in the figure, the electrochemical potential in each segment is denoted by $\mu_1$ through $\mu_5$, from left to right.

Throughout we assume that the system's global fermion parity is preserved exactly. In practice, parity conservation can be enforced by operating the qubit as a 'floating' device, i.e.,

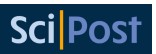

Figure 5: Qubit design considered here: the wire is separated into five segments that can be separately gated and driven in and out of the topological regime. In this illustration, the leftmost and the two rightmost regions are topological, while two regions in the middle are trivial, leading to four localized Majorana zero modes.

at most weakly coupled to leads connecting it to ground. The qubit would then exhibit a finite charging energy, which suppresses quasiparticle poisoning from non-equilibrium quasiparticles in the leads [26, 27]. Since a charging-energy term makes simulations far more challenging, however, we do not include its effects explicitly here.

When configured as a qubit, the system contains at any time a total of four MZMs—one at each end, plus two adjacent to a non-topological region within the wire—leading to a four-fold ground-state degeneracy. Within a given fixed global-fermion-parity sector, only two ground states are available, which furnish the qubit's computational states.

It is important to note the limitations of this strictly one-dimensional qubit design: While the computational states originate from a topological degeneracy, topological qubit operations (i.e., by braiding [4] or measurement-only topological quantum computation [43]) are not available. Instead qubit rotations proceed by selectively breaking the topological degeneracy and inducing non-universal couplings between the MZMs. Therefore, while the encoded quantum information enjoys topological protection when all four MZMs are well-separated, logical gate operations are unprotected and susceptible to noise, inaccuracies of the applied pulses etc., in the same way as conventional qubits. In this paper, we will restrict ourselves to idealized models where the only sources of errors are the finite length of the wire and diabatic corrections. For a recent discussion of other corrections, see Ref. [44].

The system's low-energy subspace can be described by the effective Hamiltonian

$$H = i \sum_{i<j} \varepsilon_{ij} \gamma_i \gamma_j. \tag{16}$$

Here $\gamma_i$ is the Majorana operator associated with the $i^{\text{th}}$ MZM numbered left to right as in Fig. 5, and $\varepsilon_{ij}$ is the coupling resulting from finite overlap of MZMs $i$ and $j$. Such a simplified description disregards diabatic excitations to states above the superconducting gap, but is nevertheless instructive for developing a simple picture of qubit operations. After fleshing out this minimalist picture we return to numerical simulations of more complete microscopic descriptions of our qubit protocols.

In the present qubit design, the $\varepsilon_{ij}$'s are tuned by changing the positions of the MZMs (Ref. [45] discussed a similar qubit where the couplings were tuned via a different mechanism). We assume throughout this discussion that the MZMs remain separated by more than a superconducting coherence length, so that the coupling energy between them can be modeled as $\varepsilon \sim \frac{v_F}{\xi} e^{-d/\xi} \cos(\kappa d)$, where $\xi$ is the coherence length of the superconductor, $d$ is the distance between the two MZMs, and $\kappa$ is on the scale of the Fermi momentum $k_F$. An expression of this form holds for both the case of MZMs separated by a topological region and a trivial region [46]; however, the relevant coherence length and the prefactor (omitted in the expression above) generally differ in the two cases.

A convenient basis for the low-energy Hilbert space is obtained by combining the left and right pairs of Majorana modes into complex fermionic operators, $f_L = (\gamma_1 + i\gamma_2)/2$ and $f_R = (\gamma_3 + i\gamma_4)/2$, and using the occupation number basis for these complex fermions. At fixed overall parity $P = (i\gamma_1\gamma_2)(i\gamma_3\gamma_4) = \pm 1$, the relevant basis states are either $\{|01\rangle, |10\rangle\}$

or $\{|00\rangle, |11\rangle\}$. Without loss of generality we assert that the system resides in the latter, even-parity sector. Upon identifying computational states $|0\rangle \equiv |00\rangle$ and $|1\rangle \equiv |11\rangle$, the Hamiltonian 16 can be reduced to

$$H = (\varepsilon_{12} + \varepsilon_{34})\sigma^z + (\varepsilon_{23} + \varepsilon_{14})\sigma^x + (\varepsilon_{13} - \varepsilon_{24})\sigma^y, \tag{17}$$

where $\sigma^{x,y,z}$ denote Pauli matrices that act on the qubit. Since the interactions between MZMs are exponentially suppressed in their separation, it is reasonable to assume that only nearest-neighbor interaction terms are relevant. In this limit, the Hamiltonian simplifies to

$$H = (\varepsilon_{12} + \varepsilon_{34})\sigma^z + \varepsilon_{23}\sigma^x. \tag{18}$$

## 3.1 Rabi oscillations

A simple yet powerful qubit protocol involves demonstrating Rabi oscillations, i.e., coherent oscillations of a two-level quantum system. Initially, the system parameters are chosen such that the $\sigma^z$ term dominates the Hamiltonian, and the system is initialized into the ground state. The $\sigma^x$ term is then increased to be the dominant coupling, causing the qubit to precess around the $x$ axis. Finally, the qubit is measured in the $z$ basis.

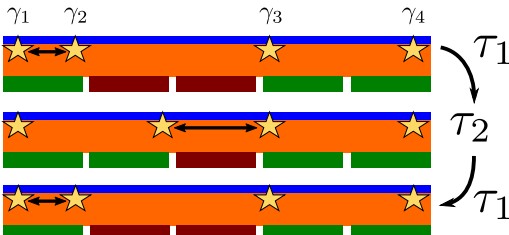

Figure 6: Basic Rabi-oscillation protocol. The qubit is initialized with the leftmost pair of MZMs close to each other, so that their coupling dominates the low-energy Hamiltonian; their joint parity can be read out for initialization. Then, over a time $\tau_1$, a piano-key move shuttles $\gamma_2$ closer to $\gamma_3$ so that their coupling dominates—causing the qubit to precess for a waiting time $\tau_2$. In the final step, a piano-key move over time $\tau_1$ returns $\gamma_2$ to its original location, whereupon the qubit state is read out. Double arrows indicate the dominant coupling for each stage.

In our qubit setup we can change the relative strength of the $\sigma^x$ and $\sigma^z$ terms by controlling which MZMs are closest to each other. Figure 6 illustrates an implementation of the above Rabi-oscillation protocol. Here, the two rightmost MZMs reside at fixed positions throughout the protocol; however, one could equally well perform a symmetric protocol where all operations are performed on both the left and right pairs of MZMs. The system is initialized with a dominant coupling $\varepsilon_{12}$. A piano-key move performed over a time $\tau_1$ transfers $\gamma_2$ closer to $\gamma_3$, yielding $\varepsilon_{23}$ as the dominant coupling. Next, we let the system evolve (oscillate) under the $\sigma^x$-dominated Hamiltonian for a time $\tau_2$, and then apply a second piano-key move over a time $\tau_1$ to revert to the original configuration. A final measurement determines the expectation value $\langle i\gamma_1\gamma_2\rangle$, i.e., the occupation of the complex fermion formed when bringing the two leftmost MZMs close to each other. Such a measurement could be performed by a nearby quantum dot [27, 47, 48]. The latter can be included by extending the nanowire to the left, outside of the region that is covered by the superconductor.

To achieve high-fidelity Rabi oscillations, the manipulations of the electric gates that transport the MZMs should occur sufficiently slowly so that diabatic corrections are minimal, i.e., $\tau_1$ should be long compared to the characteristic piano-key timescale $\tau_0$ discussed in Sec. 2.

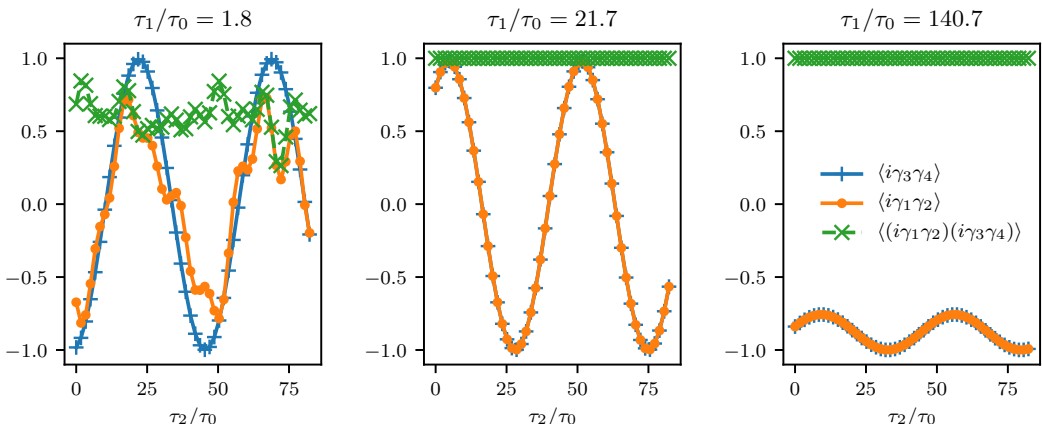

Figure 7: Representative behavior of the occupation of the left pair of Majorana zero modes, $i\gamma_1\gamma_2$, the right pair, $i\gamma_3\gamma_4$, and the parity in the low-energy subspace $-\gamma_1\gamma_2\gamma_3\gamma_4$, during the Rabi oscillation protocol sketched in Fig. 6. From left to right we show three choices of the timescale $\tau_1$ for moving the MZMs with increasing adiabaticity. Simulations are performed in a Kitaev-chain model with $\mu_{\text{top}} = 1.7$, $\mu_{\text{triv}} = 2.3$, $\Delta = 0.3$ and total system size $L = 60$.

Conversely, consider the limit where all operations are purely adiabatic. Throughout the evolution the system then follows the instantaneous ground state of the Hamiltonian—which due to splittings is generically unique—and no operations are performed on the qubit modulo an irrelevant overall dynamical phase. Therefore, gate manipulations must also be sufficiently fast compared to an energy scale which we denote as $E_{\text{s}}$. Roughly speaking, $E_{\text{s}}$ corresponds to the minimal gap of the Hamiltonian in a regime where it is dominated by a $\sigma^x$ interaction. The specific value depends on microscopic details, in particular oscillations of the splitting terms, and is hard to determine. A worst-case estimate can be given as $E_{\text{s}} = \max\{\varepsilon_{23}\}$, i.e., the maximal coupling between the middle two MZMs; in practice, the relevant energy scale will usually be smaller and the constraint thus less stringent. To summarize, the constraints on $\tau_1$ are [2]

$$\tau_0 < \tau_1 < E_{\text{s}}^{-1}. \tag{19}$$

The coupling between $\gamma_2, \gamma_3$ during the $\sigma^x$-dominated part of the protocol sets the scale for a conservative estimate for the upper limit on $\tau_1$: $E_{\text{s}} \sim \frac{v_F}{\xi} e^{-L_{\text{key}}/\xi}$, where $v_F$ and $\xi$ are parameters appropriate for MZMs hybridized across a trivial region of length $L_{\text{key}}$. Note that for this choice, the Rabi frequency $\omega$ and $E_{\text{s}}$ coincide.

As shown in Sec. 2, the characteristic time $\tau_0$ associated with diabatic corrections scales quadratically with the size of a piano key $L_{\text{key}}$ [see Eq. (9)]. The operating regime for the qubit thus increases rapidly with system size, since $E_{\text{s}}^{-1}$ increases exponentially with $L_{\text{key}}$. We will now confirm this behavior numerically by studying microscopic models that include diabatic corrections explicitly.

Figure 7 shows a simulation of the above Rabi oscillation protocol in a Kitaev-chain model. Panels represent data for different piano-key times $\tau_1$, ranging from $\tau_1/\tau_0 \sim 1$ (left) to $\tau_1/\tau_0 \gg 1$ (middle and right); see caption for other parameters. In each case we plot the occupation of the left and right MZM pairs, $\langle i\gamma_1\gamma_2 \rangle$ and $\langle i\gamma_3\gamma_4 \rangle$, as well as the ground-state fermion parity, $\langle (i\gamma_1\gamma_2)(i\gamma_3\gamma_4) \rangle$, versus $\tau_2$. As in Sec. 2.2, the $\gamma_i$ operators are defined through eigenvectors of the final Hamiltonian. It follows that eigenstates of the final Hamiltonian are

---

[2]Note that during initialization and readout, $\varepsilon_{12}$ can exceed $1/\tau_1$ without causing any issues. The inequalities in Eq. (19) apply only during the manipulation stage of the Rabi-oscillation protocol.

also eigenstates of $(i\gamma_1\gamma_2)(i\gamma_3\gamma_4)$, and that deviations from $\langle(i\gamma_1\gamma_2)(i\gamma_3\gamma_4)\rangle = 1$ in the final state indicate excitations away from the low-energy manifold.

We observe that for protocols with fast piano-key moves, i.e., $\tau_1 \sim \tau_0$, diabatic corrections indicated by $1 - \langle(i\gamma_1\gamma_2)(i\gamma_3\gamma_4)\rangle$ become sizable as seen in the left panel. Correspondingly, the occupation of the left MZM pair—on which the operations are performed—exhibits only quite noisy oscillations as a function of $\tau_2$. Interestingly, the occupation of the right MZM pair—which in this particular protocol remains static—shows cleaner oscillations. For $\tau_1 \ll \tau_0$ (not shown), oscillations in both pairs are washed out. As $\tau_1$ increases, the oscillations become cleaner, and the two pairs of MZMs exhibit very similar behavior; see middle and right panels. Finally, for very large $\tau_1$, effects of adiabaticity with respect to the residual splitting $E_s$ are felt, thereby suppressing the oscillation amplitude. As expected this regime is approached once $\tau_1 \sim \omega^{-1}$.

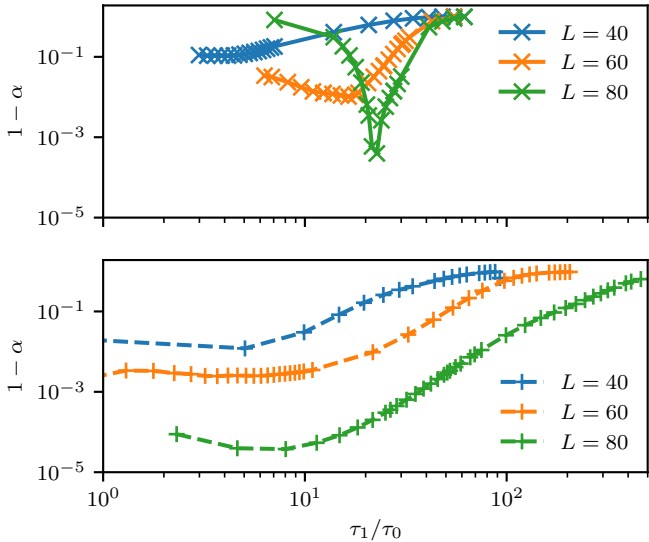

Figure 8: Optimal fidelity of a $\pi$ qubit rotation as a function of piano-key timescale $\tau_1$ obtained by fitting numerical simulations of the protocol (see Fig. 7) to Eq. (20). *Top panel:* $\Delta = 0.1$. *Bottom panel:* $\Delta = 0.3$.

We further explore the optimal operating regime for this qubit. To this end, we fit the oscillations to

$$\langle i\gamma_3\gamma_4\rangle = C_0 + \alpha\cos(\omega\tau_2 + \phi_0), \qquad (20)$$

where $C_0$, $\alpha$, $\omega$ and $\phi_0$ are all fit parameters. A similar fit can be performed for the left pair, $\langle i\gamma_1\gamma_2\rangle$, which yields comparable results away from the limit of very short $\tau_1$. The most relevant quantity is the oscillation amplitude $\alpha$, which is akin to the fidelity for a logical $X$ gate ($\pi$ qubit rotation). Since the operations discussed here are not topologically protected, $\alpha$ will approach the maximum value $\alpha = 1$ only when the qubit operation is optimized as we discuss below. It should be emphasized that this estimate includes only diabatic corrections and finite-size corrections due to undesired Majorana splittings, and does not include any external noise, effects of finite temperature, quasi-particle poisoning etc., and as such can be understood as a theoretical upper bound on the achievable gate fidelity for a given wire length. In Fig. 8, we plot $\log(1-\alpha)$ for different system sizes and over a range of protocol times for two different gap sizes in the Kitaev chain. We show only the range of data where a reliable fit to Eq. (20) can be achieved; for shorter piano-key times, the oscillations are too noisy to reliably fit the data. The results clearly demonstrate that, within this model, increasing the system size shifts the optimal operating regime to longer piano-key times, and also reduces the optimal error

that can be achieved. This trend reflects the exponential growth of the operating window specified in Eq. (19), which allows one to efficiently avoid diabatic errors and simultaneously manipulate the qubit faster than residual couplings of MZMs.

We have performed similar simulations of the Rabi protocol in the quantum-wire model of Eq. (13), and found qualitatively similar results. This allows us to estimate the minimal size of the piano key to reach the operational window in Eq. (19) in an experiment using proximitized nanowires. At the end of Sec. 2.1, we estimated that $\tau_0(L_{\text{key}}) \sim 10$ ns $\cdot (L_{\text{key}}/0.5 \ \mu\text{m})^2$. Estimating $E_s$ is more challenging, in particular since the splitting between MZMs coupled through a trivial superconductor depends on conditions that are yet not clear in practice. One can however obtain certain bounds by constraining the strength of the $\sigma^x$ term that the couples the middle two MZMs, and then using the (worst-case) estimate $E_s = \max\{\varepsilon_{23}\}$. The $\sigma^x$ term during the second step of the protocol must at the very least exceed the residual $\sigma^z$ term, which is set by the splitting through a topological region of length $2L_{\text{key}}$ in our setup. Using experimental estimates for the splitting, $\varepsilon(L) \sim 0.1$ meV $\exp[-L/(260 \text{ nm})]$ [49], we find that to satisfy $\tau_0(L_{\text{key}}) < \varepsilon^{-1}(2L_{\text{key}})$ requires $L_{\text{key}} > 850$ nm. However, for a choice of parameters where during the second step the $\sigma^x$ and $\sigma^z$ terms are comparable, the visibility of oscillations will be very limited. A more conservative choice is to consider parameters where $\sigma^x$ becomes larger, thus possibly placing tighter conditions on the length of a piano key.

It should be noted that in the regime where at all times the MZMs are well-separated, $E_s$ as well as the Rabi-oscillation frequency are exponentially sensitive to changes in $\xi$ and thus small changes of the microscopic parameters, which severely limits the reliability of numerical estimates. While the experiment could also be performed in a regime where the two middle MZMs are brought very close to each other and thus have coupling comparable to the gap, this would likely lead to Rabi-oscillation frequencies that are too high to resolve in time-domain experiments. Tuning the Rabi frequency $\omega$ into the most interesting regime where it is fast compared to decoherence times, but slow compared to the scales on which the protocols and readout can be performed, will require careful tuning of the microscopic parameters of the middle region separating the central two MZMs.

Throughout this paper we neglected decoherence processes that can arise with a finite hybridization of the Majorana modes [44]. The latter will dampen the Rabi oscillations. Note, however, that coupling of the noise to the system is limited by the strength of the Majorana hybridization, which sets the Rabi frequency. We therefore expect to be able to observe several Rabi cycles before decoherence processes take over.

## 4 Conclusions and outlook

In this paper we studied the dynamical manipulation of Majorana-based qubits by tuning a small number of electric gates (piano keys). The latter allow one to move the topological regions and their boundary MZMs. We consider the practical regime of piano keys larger than the coherence length. By explicitly simulating the time evolution of both a Kitaev chain and quantum-wire model, we show that the diabatic excitations are well-described by a Landau-Zener picture. The corresponding minimal gap is given by the finite-size level spacing of the piano-key region at criticality. For piano keys of size $L_{\text{key}}$ in a quantum wire with the parameters listed in Sec. 2.1, we estimate that adiabaticity is reached for times longer than $\tau_0 \sim 10$ ns $\cdot (L_{\text{key}}/0.5\mu\text{m})^2$. However, there is significant uncertainty in these parameters, especially the strength of spin-orbit coupling in the presence of the superconductor. The 'true' value for $\tau_0$ could easily change by one or two orders of magnitude compared to our estimate.

We then apply a single back-and-forth piano key move to perform a Rabi oscillation protocol in a topological qubit defined by four MZMs in a linear quantum wire. The simula-

tions clearly show an exponentially growing window of operation where the piano-key moves do not create unwanted excitations while still being fast compared to the residual MZM hybridization, i.e., where the constraints of Eq. 19 are satisfied. A concrete determination of the exponentially-dependent parameters is difficult but estimates for the protocol considered here indicate that a relatively large $L_{key} \gtrsim 1\,\mu$m is required to achieve sufficient separation between the residual overlap between MZMs and the time required to move the MZMs. This condition can be relaxed by using multiple piano keys during the protocol or using smart pulses [16].

A central challenge for near-term experiments is to certify that a device that is expected to host Majorana zero modes is actually topological, i.e., that the observed low-energy states are not of other origin, such as trivial Andreev bound states [50–55]. A standard approach to verify the presence of a topological phase is to check for robustness of the observed behavior against small variations of key parameters such as the magnetic field, the electrochemical potential (tunable via gating), etc.

A second, important check is that the Rabi frequency is expected to depend exponentially on the ratio of the separation between the two middle MZMs during the step where they are hybridized to the superconducting coherence length. This scaling can be tested by fabricating devices of different lengths; indeed, experimental evidence for such exponential scaling of the relevant energy scales for hybridization has been reported [49]. Alternatively, the coherence length can be changed by tuning microscopic parameters such as the electrochemical potential or magnetic field. Finally, in a device with more than the minimal number of five piano keys discussed here, one can either use only parts of the system or perform more complicated protocols to effectively probe different lengths in the same device.

It is in principle possible to form a similar qubit to the one presented here not with topological degrees of freedom, but rather just trivial low-energy states such as Andreev bound states that are accidentally tuned close to zero energy. In this scenario, instead of tuning the separation between MZMs, the gates are used to tune the energy of these trivial states. Since all operations discussed here are based on breaking topological degeneracy and are thus not topologically protected—unlike for example braiding, which would be possible in more sophisticated topological qubits—it can be difficult to distinguish these two scenarios.

There are a number of consistency checks that can be performed whose failure would indicate that the qubit is not based on MZMs; however, their success does not necessarily rule out non-topological behavior. One example is to tune the rightmost segments of the wire (far away from the readout) out of the putative topological phase. In this case, there should be no ground-state degeneracy in a fixed parity sector and thus no Rabi oscillations. Such a test determines that oscillations are not due to purely local effects near the readout. Another useful tool is to study the finite-size dependence of Rabi oscillations. While an exponential scaling of the Rabi frequency with system size is also possible in a qubit based on localized Andreev states near the ends of the system, fine-tuning these states close to zero energy should become more and more challenging as well, and the visibility of oscillations should decrease with system size. As discussed in Ref. [30], a further important consistency check is to confirm the correlation between measurement outcomes at the left and the right end. Depending on whether the overall parity is even or odd, the measurements should either be correlated or anti-correlated, i.e. $\langle i\gamma_1\gamma_2 \rangle = \pm \langle i\gamma_3\gamma_4 \rangle$.

A powerful way to assert that a qubit is indeed based on topological degrees of freedom is to study the behavior of the system in the presence of low-frequency local noise. Some signatures in this context were discussed in Ref. [45]; extensions of these ideas to qubit designs as considered here will be discussed in future work [56].

## 5 Acknowledgements

We thank P. Schmitteckert, S. Plugge, and M. Endres for useful discussions. JA gratefully acknowledges partial support from the National Science Foundation through grant DMR-1723367 and the Army Research Office under Grant Award W911NF-17-1-0323. This research was also supported by the Caltech Institute for Quantum Information and Matter, an NSF Physics Frontiers Center with support of the Gordon and Betty Moore Foundation through Grant GBMF1250, and the Walter Burke Institute for Theoretical Physics at Caltech (RVM and JA).

## A  Lattice Hamiltonian for the quantum wire

For the sake of completeness, we list below the lattice Hamiltonian we use in our simulations of the quantum wire, which is described by Eq. (13):

$$H_{\text{QW}} = -\frac{t}{2} \sum_{i\sigma} \left( c^\dagger_{i,\sigma} c_{i+1,\sigma} + c^\dagger_{i+1,\sigma} c_{i,\sigma} \right) \tag{21}$$

$$+ (t - \mu) \sum_{i\sigma} c^\dagger_{i,\sigma} c_{i,\sigma} + V_z \sum_{i\sigma\sigma'} c^\dagger_{i,\sigma} (\sigma^z)_{\sigma\sigma'} c_{i,\sigma'} \tag{22}$$

$$+ \frac{\alpha}{2} \sum_{i\sigma\sigma'} \left( c^\dagger_{i,\sigma} (i\sigma^y)_{\sigma\sigma'} c_{i+1,\sigma'} + c^\dagger_{i+1,\sigma'} (i\sigma^y)_{\sigma\sigma'} c_{i,\sigma} \right) \tag{23}$$

$$+ \Delta \sum_{i} \left( c_{i,\uparrow} c_{i,\downarrow} + c^\dagger_{i,\downarrow} c^\dagger_{i,\uparrow} \right). \tag{24}$$

Here, we set the unit of energy as $t = 1$, and all other parameters are as described following Eq. (13).

## B  Time evolution formalism

We consider a Hamiltonian of the form

$$H = \frac{i}{4} \sum_{i,j=1}^{N} A_{ij} a_i a_j, \tag{25}$$

where $a_i$ are Majorana fermions with $\{a_i, a_j\} = 2\delta_{ij}$, $a^\dagger_i = a_i$, $(a_i)^2 = 1$, and $A_{ij}$ is skew-symmetric, $A_{ij} = -A_{ji}$. Any quadratic fermionic Hamiltonian can be brought into this form. The matrix $A$ can be brought into a block-diagonal "canonical form" of $2 \times 2$ blocks, $B = \bigoplus_{k=1}^{N/2} \varepsilon_k i\sigma_y$ where $\varepsilon_k$ are the non-negative eigenvalues of $iA_{ij}$ [5] and $\sigma_y$ denotes the Pauli matrix. To achieve this numerically as well as to compute the Pfaffians mentioned below, we use the software package described in Ref. [57]. The Hamiltonian then takes the form

$$H = \frac{i}{2} \sum_{k=1}^{N/2} \varepsilon_k \gamma_{2k-1} \gamma_{2k}. \tag{26}$$

The system can be completely described by the covariance matrix (for a more detailed description of this formalism see, e.g., Ref. [58]):

$$M_{ij} = \frac{-i}{2} \langle [a_i, a_j] \rangle. \tag{27}$$

$M$ is real and skew-symmetric, and $M^2 = -1$. To compute the covariance matrix for an eigenstate of $H$, let $O$ be the orthogonal matrix that brings $A$ into the canonical form $B$, i.e., $B = O^T A O$. Then, $M = O M_0 O^T$, where $M_0 = \bigoplus_{k=1}^{N/2} i\sigma_y$, is the covariance matrix of the ground state of $H$. More generally, the covariance matrix of an occupation number vector $|y\rangle = |y_1 \dots y_{N/2}\rangle$, $y_k \in \{0, 1\}$ in the eigenbasis is given by $M_y = \bigoplus_{k=1}^{N/2} (-1)^{y_k} i\sigma_y$, and thus $O M_y O^T$ is the covariance matrix of a (possibly excited) eigenstate with the corresponding occupation of eigenmodes of the Hamiltonian.

In terms of the covariance matrix, the time-dependent Schrödinger equation takes the form

$$\frac{dM}{dt} = [A, M]. \tag{28}$$

To perform the time evolution, we can either integrate 28 directly using a standard ODE solver, or we can (assuming that $A$ is independent of time) formally integrate it to find $M(t) = e^{At} M(0) e^{-At}$. If $A(t)$ depends on time, the time evolution can be approximated by taking it to be piecewise constant over a time step $dt$, and integrate separately over each $dt$. In that case, care must be taken that $dt$ is small enough.

Wick's theorem can be used to evaluate the expectation value of any monomial of fermionic operators as

$$\langle a_{i_1} a_{i_2} \dots \dots a_{i_n} \rangle = \mathrm{Pf}\left( i M_{i_1 \dots i_n} \right), \tag{29}$$

where $\mathrm{Pf}(\cdot)$ denotes the Pfaffian of the matrix, and $M_{i_1 \dots i_n}$ the restriction of $M$ onto the (Majorana) "sites" $i_1, \dots, i_n$. In particular, the total fermionic parity of the state corresponding to $M$ is given by $\mathrm{Pf}(iM)$. Furthermore, the modulus of the overlap of two wavefunctions is given by

$$|\langle \phi | \psi \rangle|^2 = \left| 2^{-N/2} \mathrm{Pf}(M_\phi + M_\psi) \right|, \tag{30}$$

where $M_\phi$ and $M_\psi$ are the covariance matrices corresponding to the respective wavefunctions, and $N$ is the total number of (Majorana) fermionic modes. For a discussion on how to compute the overlap including the phase, see Ref. [58].

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
