# Peer review of "Dynamics of Majorana-based qubits operated with an array of tunable gates"

_SciPost Physics, doi:SciPost Phys. 5, 004 (2018)_

## Round 1 · Referee Report · Thomas O'Brien (Referee 1) · 2018-4-28

Strengths

1- Studies an experimentally-relevant Majorana experiment.
2- Derives speed limit for adiabatic pulses with physically relevant parameters.
3- Describes and studies a small-scale Rabi experiment that appears achievable in the lab.

Weaknesses

1- Could use some estimation of how the studied errors compare to other known sources of error.

Report

The authors consider the effect of non-adiabaticity in proposals for Majorana-based qubits where individual Majorana-zero modes are physically moved by tuning underlying gate voltages. Such operations require tuning local regions between topologically trivial and non-trivial phases, which may leave residual excitations above the ground state. The authors map this tuning to a Landau-Zener problem, allowing them to calculate a critical time-scale for adiabaticity. They compare this timescale to numerical calculations of the diabatic error in the operation of two models, finding good agreement to both. Finally, the authors use these calculations to determine the diabatic contribution to errors in a Rabi pulse experiment.

The paper is very well-written, mathematically sound, and topical given the current push towards low-cost demonstrations of Majorana-based qubit devices. The authors have gone to quite great depths comparing their analytic results to numerics, and the agreement here is impressive. Though previous studies of the diabatic error in braiding experiments have been performed (Refs 18-22 of the authors), none have studied this particular system, nor the Rabi experiment proposed. Also, the explicit timescales and fidelities specified are of great use for pinning down potential error rates in Majorana qubits.

As an aside, the lack of noise in $\langle i\gamma_3\gamma_4\rangle$ in the left panel of Fig.7 can be attributed to the fact that the noise process is coherent - we know that the $|00\rangle$ state shifts to the $|10\rangle$ state if an excitation occurs, and the $|00\rangle-|11\rangle$ and $|01\rangle-|10\rangle$ gaps are the same (assuming $\epsilon_{14},\epsilon_{13},\epsilon_{24}<<\epsilon_{23}$), so no dephasing occurs. This implies that the additional decay from the excited state will affect this coherence fairly heavily (which will not be the case for the other two panels where the system is practically always in the low-energy subspace).

Requested changes

1- I can't find the units for $\tau_2$ in Fig.7 (and similarly in Fig.8 for $\tau_1$). I assume both are $\tau_0$?
2- The oscillation frequency in Fig.7 is dictated by $\epsilon_{23}$, for which an approximation can be found at the top of page 6. Currently this is seemingly chosen to be rather small (~200kHz assuming $\tau_2$ is in units of $\tau_0\approx 10$ ns). Could the authors comment on how large this could be made?
3- It would be helpful if the authors included a comparison to known timescales of other noise processes in Majorana qubits. This would allow for a (very tentative) calculation of the number of braiding steps that could performed on Majorana qubits before these other processes took hold, and would be of use in comparing to the fidelities given in Fig.8.
4- Could the authors compare their determined timescales with those found in Refs 18-22, or at least comment on how applicable they expect their $\tau_0\approx 10$ ns value to be to other Majorana-based schemes?

  • validity: high
  • significance: high
  • originality: good
  • clarity: top
  • formatting: excellent
  • grammar: excellent

Author:  Torsten Karzig  on 2018-06-02  [id 265]

(in reply to Report 1 by Thomas O'Brien on 2018-04-28)
Category:
answer to question

We thank the referee for his review and the comment regarding the noise of $\langle i\gamma_3\gamma_4\rangle$.

Regarding the other noise sources (question 3), we added a short paragraph to the end of Sec. IIIA of the paper. It should be possible to isolate the system sufficiently from the environment to neglect external quasiparticle poisoning. The remaining error sources are thermal excitations and noise coupling to the Majorana system once there is a finite hybridization (for example throughout the Rabi oscillation process). At low temperatures the latter noise is dominating. The coupling to the noise, however, is limited by the size of the hybridization coupling that is controlling the Rabi frequency. For typical (charge) noise Ref. [45] estimates that the noise will only weakly perturb the hybridization coupling so several Rabi cycles should be observable.

Below we reply to the remaining questions:

1 -- The time scales in Fig. 7 and 8 were actually plotted in different units. They are now reported in terms of $\tau_0$.

2 -- Note that with the correct units of Figs. 7 and 8 an estimate for the corresponding oscillation frequency would be $\sim 2$ MHz. We expect these scales to be reachable due to the exponentially small hybridization of the Majorana modes. Moreover, the scheme would work similarly if the oscillation frequency becomes of the order of $\tau_1$.

4 -- We added a comment to the paper comparing the time scales to the approaches in Refs 18-22. In general one would expect that the conditions for adiabaticity are more stringent in an approach with large piano keys since the level spacing at the critical point is smaller than typical values of the topological gap in Refs 18-22.

---

## Round 1 · Referee Report · Anonymous (Referee 3) · 2018-5-2

Strengths

1- Authors study a relevant model for diabatic corrections when shuttling Majorana fermions
2- The Landau-Zener approximate result is compared to numerics and appears convincing
3- The effect on non-topological qubit operation is extracted

Weaknesses

1 - There is no discussion of open system dynamics when the Majorana fermions are hybridized
2 - There is little discussion on how these results could be measured

Report

The Authors study the diabatic excitations associated with previous proposals for shuttling Majorana fermions using electric gates. They find that they are well described by the Landau-Zener formula, and estimate the condition for adiabaticity. They consider a protocol to generate Rabi oscillations in a qubit with four Majorana fermions, by shuttling a single Majorana zero mode back and forth.

The paper is timely and well-written. The results seem sound and consistent. The paper contributes to the study of diabatic errors with Majorana fermions by considering a relevant protocol.

Requested changes

1 - The values for the various parameters in the figures and axis labels should be specified with units
2- Adding a discussion regarding the measurement procedure may be useful
3 - There is a typo in the conclusions, "magnetid" -> "magnetic"

  • validity: high
  • significance: good
  • originality: good
  • clarity: top
  • formatting: excellent
  • grammar: excellent

Author:  Torsten Karzig  on 2018-06-02  [id 264]

(in reply to Report 2 on 2018-05-02)
Category:
answer to question
reply to objection

We thank the referee for his/her comments which we address below.

-- We work in the limit where the system is sufficiently isolated from the environment to prevent quasiparticle poisoning. We expect other noise sources, such as the dephasing discussed in Ref. [45] (Knapp et al, PRB 97, 125404 (2018)), to be subleading during the short times a Rabi protocol would be performed. We added a corresponding comment (see also reply to referee I).
-- The Majorana parity $i\gamma_1 \gamma_2$ could be read out by a nearby quantum dot. The latter can be included by extending the quantum wire beyond the region that is covered by the superconductor. For a discussion, see for example the new Ref. [31] of the manuscript (Gharavi et al, PRB 94, 155417 (2016)).

---

## Round 1 · Referee Report · Anonymous (Referee 2) · 2018-5-5

Strengths

The authors of the manuscript titled
"Dynamics of Majorana-based qubits operated with an array of tunable gates" have studied
explicitly the dynamics of Majorana wires in a variant of the protocol in Ref 13 where
the gate segments is sufficiently long to support reasonably weakly split Majorana modes.
Specifically the authors study the possibility of diabatic errors from excitations
of quasiparticles in these relatively long gated wires and determine parameters such as
how fast can MZMs be moved in these geometries relative to splitting-induced qubit dephasing.
This is used to determine parameters where Rabi oscillations may be seen in the simplest
system. One of the key results is that if the rate is slower than a scale that goes
inversely as the square of the length of the system then the MZM transfer process can be
performed essentially perfectly. This is consistent with the expectation from Kibble Zurek
(i.e. defect density ~ 1/sqrt(tau)) (see Phys. Rev. B 85, 165425 (2012)) for example.
However, what is surprising, at least from consideration of previous numerical results
(Refs. 20 and 24) is that the error from Figs 2,3 appears to be exponentially small.
If generically true in these nanowire devices, this would be a rather encouraging answer
to some of the concerns of diabatic errors discussed in Ref. 20. The scaling here is
rather important for any claim of genericness of the answers since the authors themselves
admit in the phrase "Some caution is warranted with such estimates, however."

Weaknesses

Therefore before recommending publication I have a few suggestions along the lines of
establishing the genericness of the scaling behavior of the diabatic errors claimed
in Figs. 2,3. These are:

(1) The analytic argument based on the usual Landau Zener formula (Eq. 7) seems dubious
to me. My main worry here is that this the references cited establish this for analytic
dependences of the mass delta mu(t). On the other hand, the present propotol starts and
ends at a finite point in time. While the function has a continuous derviative as opposed
to Ref 20, it is still not analytic at the end points. According to
G. A. Hagedorn and A. Joye, Journal of mathematical analysis
and applications 267, 235 (2002), this can be a major issue. The authors should
clarify whether they strictly expect exponential scaling suggested by Eq. 7 or not and
if they do why given that the references they suggest here do not apply.

(2) At some level the numerical results in Fig 2,3 over-ride 1 at least empirically - but
then it becomes a concern whether the exponential behavior seen in 2,3 is some artifact of
the model studied in the paper. One technical point here is what is the value of a to be
used in Eq. 14. From appendix A, a looks like an arbitrary lattice parameter that should
not appear in the nanowire equation. The authors should specify how a is to be selected in
the paragraph below Eq. 14 where they talk about parameters. More seriously, in the absense
of a generic argument, the authors should account for some more potential deviations such as
barrier potentials and barrier dynamics that might arise at the interface from the dynamics
of the gate voltage a screening potential. It is possible that these impurity and
interface dynamics breaks the exponential dependence of the error seen in Fig.2,3

(3) The authors should discuss why Ref 20 sees power-law diabatic errors and
the present manuscript finds exponential. This could probably be simply achieved by
comparing the different choices for the function f(s) that is used to set the time
dependence of the gate potential.

(4) The authors make a few comments about comparison to some existing proposals
with Coulomb blockade for manipulation of MZMs. The calculation in the present paper have
no such interaction effects. It is unclear to me how these are related or which one is
superior.

(5) The authors cite "improvements to gating techniques" as a motivation for the present
work. It would be helpful if the authors could be more specific e.g. provide a reference.

Report

In summary, I think the present manuscript examines an important question of whether
time variations of the gate potential are a viable route to qubit manipulation with
Majorana zero modes. However, the general scaling of the time claimed is consistent with
traditional Landau Zener arguments. So the real interest is whether a reliable argument is
made for the scaling of the error is really exponential for reasonable device models. I
feel I must withhold my recommendation of this manuscript as an outstanding work
that meets the standard for publication in Scipost until the genericness is established.

Requested changes

Please see weaknesses.

  • validity: high
  • significance: high
  • originality: good
  • clarity: high
  • formatting: perfect
  • grammar: perfect

Author:  Torsten Karzig  on 2018-06-02  [id 263]

(in reply to Report 3 on 2018-05-05)
Category:
answer to question
reply to objection

We thank the referee for his/her comments. The referee's concerns focus on the issue whether for long times one would expect an exponential or power law dependence of non-adiabatic errors with the duration of the protocol. Indeed, since the protocol we use for the time evolution is not analytic one would expect a power law for long times. Importantly, however, the prefactor of the power law is parametrically suppressed in the ratio of the smallest and the final gap of the Landau-Zener process. Since the smallest gap is given by the level spacing of the "piano-key region" at the critical point, this ratio is small. Furthermore, since the discontinuity only appears in the second derivative in our protocol, this ratio enters at a high power. We derive and estimate for the prefactor of the power law below.

The amplitude $A_m$ for non-adiabatic excitations for a time-evolution with a discontinuity in the $m$'th derivative can be estimated by $m$'th order perturbation theory (see, e.g. [R1]) to be
\[
A_m=\frac{\partial_t^m h(t)|_{t=t_0+\eta}-\partial_t^m h(t)|_{t=t_0-\eta}}{E_\text{G}^{m+1}}\Bigg|_{\eta \to 0}\,,
\]
where $h(t)$ is the matrix element of the Hamiltonian with the jump in the $m$'th derivative at $t=t_0$ and $E_\text{G}$ is the gap protecting non-adiabatic transitions at $t_0$. In our case, we can estimate $A_2$ by using $h(t)=\Delta \mu f(t/\tau)$. For $f(t/\tau)=\sin^2(\frac{\pi}{2} \frac{t}{\tau})$ we then find
\[
A_2=\left(\frac{\pi}{2}\right)^2\frac{\Delta \mu}{E_\text{G}^3 \tau_0^2}\left(\frac{\tau}{\tau_0}\right)^{-2}=\left(\frac{\pi}{2}\right)^2\frac{\delta \varepsilon^4}{ E_\text{G}^3 \Delta \mu}\left(\frac{\tau}{\tau_0}\right)^{-2}\,,
\]
where $\tau_0=\Delta \mu/ \delta\varepsilon^2$ is the characteristic timescale of adiabaticity that is controlled by the minimal level spacing $\delta \varepsilon$. Typical parameters of our simulation are $\Delta\sim \Delta \mu \sim E_\text{G}$ and the smallest total system size used in Fig. 2 is $L=60$. This gives an upper bound for the prefactor $A_2\sim 3\cdot 10^{-4} (\tau/\tau_0)^{-2}$. Since Fig. 2 plots the squared amplitude $A_2^2\sim 10^{-7} (\tau/\tau_0)^{-4}$ we do not expect the power law dependence to be visible at the plotted scales. It also shows that our exponential estimate is quite accurate in the describing the onset of the adiabatic regime which is most relevant for experimental implementations.

To further test the above argument we performed a simulation with a Hamiltonian where the first derivative jumps at $t=0$, $f(t/\tau)=\Delta \mu t/\tau$. See attached figure where we used the parameters $\Delta \mu=\Delta=0.5$. We indeed observe that the exponential gets saturated by a $\tau^{-2}$ power law for long times. The onset of the power law is consistent with the above argument. Using again, $\Delta\sim \Delta \mu \sim E_\text{G}$ with $L=60$ yields $A_1^2\sim 10^{-4}(\tau/\tau_0)^{-2}$, which agrees with the onset of the power law in the simulation.

We added a comment to the paper that the exponential behavior will eventually be cut off by a power law at long times.

Finally we comment on the remaining points of the referee:

-- There should be no lattice parameter $a$ in Eq. (14). Thank you for spotting the typo.

-- To clarify question (4): There are proposals [26,27] that use the presence of charging energy to protect the qubits from quasiparticle poisoning while other proposals [25] use charging energy to create a selective coupling between MZMs. These approaches are quite different from the piano-key approach where MZMs are moved physically. Proposals [26,27] rely on a measurement-only approach while proposal [25] changes couplings between fixed MZMs. Our current understanding is that the measurement based schemes are the most promising way towards scalable qubit architectures. Nevertheless, the piano-key approach might be easier to implement for earlier proof-of-principle experiments as proposed in this paper.

-- We provided 2 references [28,29] addressing (5).

[R1] L.M. Garrido, F.J. Sancho, Physica 28, 553 (1962).

Attachment:

powerlaw.pdf

---

## Round 2 · Author Response

Minor modifications and clarifications addressing the referee reports. See list of changes and referee replies for details.

---

## Round 2 · List of Changes

• Added references [28,29].
  • Typo corrected in Eq. (14).
  • Comment added to the end of Sec. II.A comparing the non-adiabatic time scales $\tau_0$ with other schemes [18-22].
  • Comment added to Sec. II.B mentioning the eventual crossover from exponential to power-law behavior of the non-adiabatic error with time.
  • Units added to the time scales $\tau_2$ in Figs. 7 and 8.
  • Paragraph added to the end of Sec. III.A commenting on decoherence processes.

---

## Editorial Decision

published